# WSI-BayesUNet: Uncertainty-Aware Deep Learning for Histopathological Image Segmentation with Active Learning

**Yijun Cui**                                           Yijun.Cui@radboudumc.nl
*Computational Pathology*
*Radboud UMC, Netherlands*

**Geert, Litjens**                                geert.litjens@radboudumc.nl
*Computational Pathology*
*Radboud UMC, Netherlands*

**Khalili, Nadieh**                             Nadieh.Khalili@radboudumc.nl
*Computational Pathology*
*Radboud UMC, Netherlands*

## Abstract

Histopathological image segmentation is a core task in digital pathology, supporting applications such as cancer detection and subtype classification. Manual annotation is time-consuming and subjective, making automation essential for improving efficiency and consistency in diagnostic workflows. Although deep learning models have significantly automated this process, they still make silent mistakes. Quantifying the uncertainty of the model and using the uncertainty for further improvement is not fully addressed. The most common way to quantify uncertainty is through ensemble methods, which provide empirical uncertainty estimation but face limitations, including high computational costs and theoretical instability. To address these, we propose a Bayesian U-Net framework that employs variational inference for principled probabilistic uncertainty estimation. Leveraging active learning, our Bayesian U-Net iteratively improves segmentation performance by prioritizing the most uncertain samples. Experiments on the TIGER and CAMELYON17 datasets show that Bayesian U-Net outperforms ensemble methods, offering better uncertainty quantification, uncertainty-guided performance gains, and faster convergence. Notably, uncertainty-based sampling consistently surpasses random sampling, significantly reducing annotation effort while maintaining or improving segmentation accuracy. **Keywords:** Histopathology, Segmentation, Deep Learning, Uncertainty Quantification

## 1 Introduction

Histopathological images are the gold standard in diagnosis, providing vital insights into tissue architecture, cellular morphology, and disease-specific patterns. Accurate segmentation of key anatomical structures, such as invasive tumors and tumor-associated stroma, is essential for effective disease classification and prognosis. However, manual segmentation is highly time-consuming and prone to inter-observer variability, particularly due to the high resolution and complexity of histopathological images. This underscores the critical need for automated, consistent, and reliable segmentation methods.

Deep learning (DL), particularly convolutional neural networks like U-Net, has dramatically advanced segmentation automation Van der Laak et al. (2021), improving accuracy, standardizing

diagnostic workflows, and alleviating the manual workload. Nevertheless, predictive uncertainties in DL predictions present significant challenges Abdar and et al. (2021). These uncertainties primarily stem from aleatoric uncertainty, associated with inherent data ambiguity, and epistemic uncertainty, resulting from insufficient or inadequately representative training data Hüllermeier and Waegeman (2021). Reliable quantification of predictive uncertainties is vital for clinical application, enabling pathologists to identify low-confidence predictions and prioritize cases for review effectively.

Current methods for quantifying predictive uncertainty, such as ensemble approaches, provide empirical estimates through inter-model variance but suffer from significant limitations including high computational costs and theoretical instability Lakshminarayanan et al. (2017); Khalili et al. (2024). To address these issues, we propose a Bayesian deep learning framework specifically designed for quantifying predictive uncertainty in histopathological segmentation. Bayesian methods deliver principled epistemic uncertainty estimates by modeling network parameters as probability distributions, allowing efficient probabilistic inference Blundell and et al. (2015); Gal and Ghahramani (2016). Additionally, we integrate active learning Settles (2009), iteratively selecting the most uncertain samples to optimize annotation efficiency.

The primary contributions of our work include the development of WSI-BayesUNet, a Bayesian adaptation of the nnU-Net framework using variational inference for robust uncertainty quantification on whole-slide images. We use the quantified uncertainty as an active learning signal, allowing the model to identify and correct its uncertain predictions, thereby optimizing annotation efficiency and improving segmentation accuracy. Comprehensive evaluations on the TIGER and CAMELYON17 datasets demonstrate that WSI-BayesUNet surpasses ensemble-based approaches in both uncertainty estimation and segmentation performance.

## 2 Datasets

The TIGER dataset (TIGER Challenge Organizers (2021)) comprises whole-slide images (WSIs) from patients with Her2+ and Triple-Negative Breast Cancer (TNBC), obtained from multiple institutions including TCGA, Radboud University Medical Center (RUMC). Our segmentation targets invasive tumors and associated stromal tissues, critical regions for tumor-infiltrating lymphocyte assessment and diagnostic relevance.

The areas with tissue segmentation of images from the TIGER dataset are tessellated into 4,290 image patches, with 214 patches from a subset of patients used for pre-training, 1,930 patches from different patients held out for validation, and the remaining 2,146 patches—drawn from all patients—forming a pool from which our active-learning routine iteratively queried the most uncertain samples.

The CAMELYON17 dataset (CAMELYON Challenge Organizers (2017)) includes hematoxylin and eosin (H&E) stained WSIs of lymph node sections from five Dutch medical centers. Our objective is to segment metastatic lesions, which are vital for clinical staging and prognosis.

Tessellating yields 10,396 patches: 311 patches from a subset of patients are allocated for pre-training, 2,806 patches from different patients are used for validation, and the remaining 7,279 patches—drawn from all patients—form the pool from which active learning selects the most un-

certain samples.

## 3 Methods

### 3.1 Bayesian UNet

Bayesian neural networks (BNN) integrate principal epistemic uncertainty estimates into model predictions by treating the network's weights as random variables with a prior distribution $p(\mathbf{w} \mid \mathcal{D})$ rather than fixed parameters. BNNs enable the quantification of uncertainty in the network weights' inaccuracies and propagate this uncertainty through the network to the network output, providing valuable information on predictive uncertainty.

Bayesian neural networks use variational inference to approximate the posterior distribution of the model parameters. Given a data set $\mathcal{D}$, we aim to estimate the posterior distribution $p(\mathbf{w}|\mathcal{D})$, which is generally intractable. Variational inference addresses this by introducing a simpler, parameterized distribution $q_\theta(\mathbf{w})$, optimized to approximate $p(\mathbf{w}|\mathcal{D})$. The optimization objective is the Evidence Lower Bound (ELBO): $\text{ELBO}(\theta) = \mathbb{E}q_\theta(\mathbf{w})\left[\log p(\mathcal{D}|\mathbf{w})\right] - \text{KL}(q_\theta(\mathbf{w})\|p(\mathbf{w}))$, where the first term encourages the fit of the data, and the second regularizes $q_\theta(\mathbf{w})$ towards the prior distribution $p(\mathbf{w})$. Minimizing the negative ELBO yields the loss of variational inference: $\mathcal{L}\text{VI}(\theta) = -\mathbb{E}q_\theta(\mathbf{w})\left[\log p(\mathcal{D}|\mathbf{w})\right] + \text{KL}(q_\theta(\mathbf{w})\|p(\mathbf{w}))$. This formulation enables efficient Bayesian training via gradient descent and induces the accurate approximation of $p(\mathbf{w}|\mathcal{D})$ as well as fitting a model to the dataset.

In addition to the Bayesian model, for each dataset we also train an ensemble of five extended U-Nets—symmetric encoder-decoder models with skip-connected down/up-sampling blocks (Rahaman et al. (2021)) with different model's parameters initialization—using nnU-Net's augmentation and Dice-Cross-Entropy loss. The inter-model disagreement in this ensemble provides a non-Bayesian estimate of epistemic uncertainty, enabling direct comparison with the BNN-based uncertainty.

In our Bayesian deep learning framework, the data fitting term are same as the deterministic UNet loss, the regularization term are scaled down to the same order as the data fitting term due to the huge amount of network's weight and the aggregation of everyone of them.

The Bayesian model architecture is the same as the 2D U-Net architecture we use, but with the network's weights represented by a Gaussian distribution parameterized by mean ($\mu$) and log-standard-deviation ($\log \sigma$). The prior distribution of the regularization term $p(\mathbf{w})$ is a standard Gaussian distribution.

### 3.2 Predictive Uncertainty Quantification

The ensemble of five UNets, combined with the probabilistic nature of the Bayesian neural network (BNN) weights, enables the quantification of predictive uncertainty in segmentation outputs. For each input image x, predictions from five sampled models or five separately trained UNets are averaged to obtain the mean prediction $\mathbb{E}[\hat{y}]$. Uncertainty is quantified using KL divergence between individual predictions and the mean: $\text{Uncertainty}(x) = \frac{1}{5}\sum_{s=1}^{5}\sum_c \hat{y}s, c\log\frac{\hat{y}_{s,c}}{\mathbb{E}[\hat{y}]_c}$. This computation is performed pixel-wise to produce uncertainty maps. We use five Monte Carlo samples for the

BNN (sample 5 sets of weights from the network's weight distribution) and five models for the UNet ensemble to enable direct comparison as proposed and recommended by the very first work of deep ensemble on the uncertainty quantification (Lakshminarayanan et al. (2017))

### 3.3 Active Learning

To assess whether our quantified predictive uncertainty is meaningful, despite lacking ground truth of the uncertainty, we apply an uncertainty-based active learning framework to improve segmentation performance efficiently.

Instead of traditional manual annotation, we simulate active learning by using existing labels in the dataset. Our pipeline includes: 1. Initial Training: Train a Bayesian U-Net and 5 UNets on a small labeled set to seed the process. 2. Uncertainty Estimation: Estimate uncertainty on the candidate dataset pool. An uncertainty score is computed for each image by averaging KL divergence map. Select the most uncertain samples with the number same as the size of the initial small dataset. 3. Simulated Annotation & Update: Add selected samples and their ground-truth labels to the training set, then retrain the model (Both BNN and ensemble UNets). 4. Uncertainty Validation: Compare segmentation performance (Dice score) of uncertainty-based selection vs. random sampling. 5. Iteration: Repeat the uncertainty assessment and selection every 10 epochs until convergence is achieved.

This approach enables us to assess whether predictive uncertainty effectively informs annotation prioritization.

## 4 Results

### 4.1 TIGER

We followed by the proposed method described in the Method section. We pre-train ensemble nn-UNets and BayesUNe on 214 image patches pre-training dataset, we evaluate the model on validation dataset with 1930 image patches. After the pretraining, we implement the active learning approach we proposed to iteratively select the uncertain sample from a dataset pool with 2146 image patches. From the table 4.1, Uncertainty sampling benefits both models, but the Bayesian

| Model | Pretrained | 25% added | 50% added | 100% added |
|---|---|---|---|---|
| Bayesian UNet (Uncertainty Sampling) | 0.704 | 0.787 | 0.794 | 0.807 |
| Bayesian UNet (Random Sampling) | 0.704 | 0.726 | 0.753 | 0.795 |
| Ensemble UNet (Uncertainty Sampling) | 0.720 | 0.726 | 0.781 | 0.813 |
| Ensemble UNet (Random Sampling) | 0.720 | 0.743 | 0.749 | 0.801 |

Table 1: **Dice segmentation scores on Tiger** The model performance on the pretraining dataset and after we added 25%, 50%, and 100% of the remaining pool data. Uncertainty sampling lets both architectures reach 0.79 Dice with only 25%–50% extra data, while random sampling needs the 100% dataset to approach the same performance.

UNet reaps the larger gains. With a common pretrained baseline of 0.704 Dice, the Bayesian UNet jumps to 0.7866 after adding the 25 % most uncertain patches—an improvement of +8.3 percentage points(pp) over the baseline and +6.1 pp over its random-sampling counterpart (0.726). The advantage remains at 50 % data (+4.1 pp over random) and 100 % data (+1.2 pp). For the Ensemble UNet, uncertainty selection also overtakes random sampling, but only once at least half of the pool is acquired: it reaches 0.7814 and 0.8133 Dice at 50 % and 100 % data, outperforming the random baseline by around 3.2 pp and 1.2 pp, respectively, and showing a smaller overall margin than the Bayesian model as shown in the segmentation case Figure 4 This indicates that Bayesian predictive uncertainty is a more informative acquisition signal than ensemble disagreement for this task.

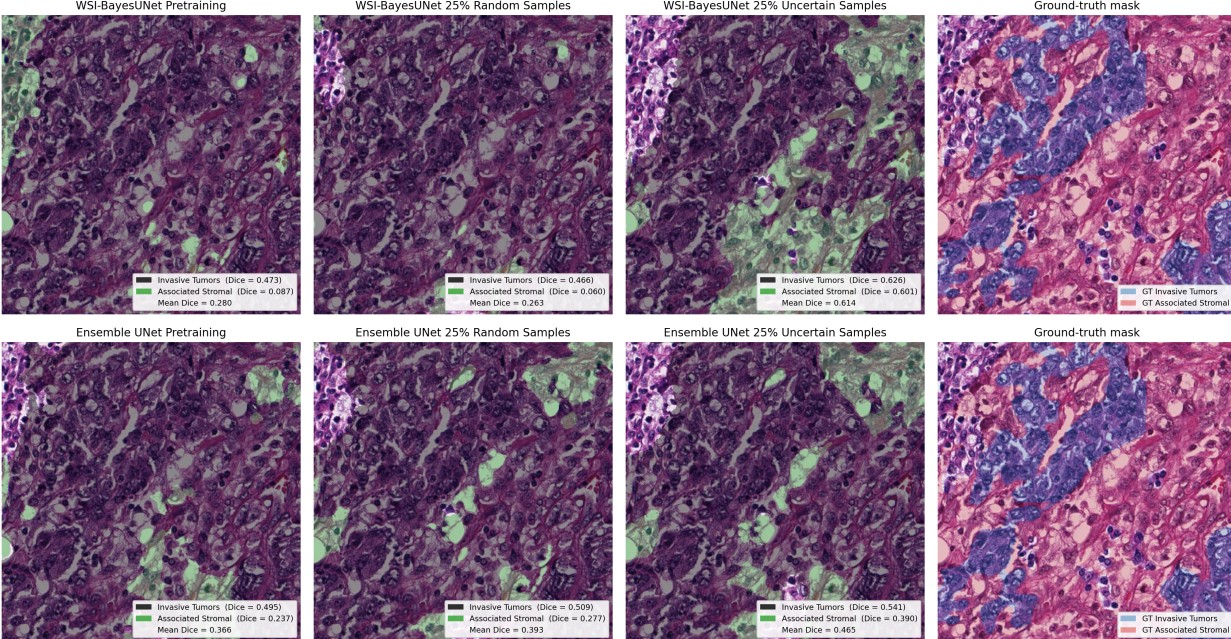

Figure 1: **Segmentation Instance FROM TIGER** Our WSI-BayesUNet improved the pretrained model on a area that are difficult to segment. The WSI-BayesUNet uncertainty sampling improved pretraining model by 119pp while random sampling didn't improve the pretraining model. Ensemble UNet uncertainty sampling only improved the pretraining model by 27pp, while the random sampling only improved the pretraining model by 6pp.

In figure 2, Bayesian uncertainty selection consistently yields a significantly higher Dice coefficient throughout training. It reaches peak performance earlier, demonstrating faster and more data-efficient learning than random sampling. While ensemble-based uncertainty selection also outperforms random sampling, it lags behind the Bayesian UNet in both convergence speed and overall performance.

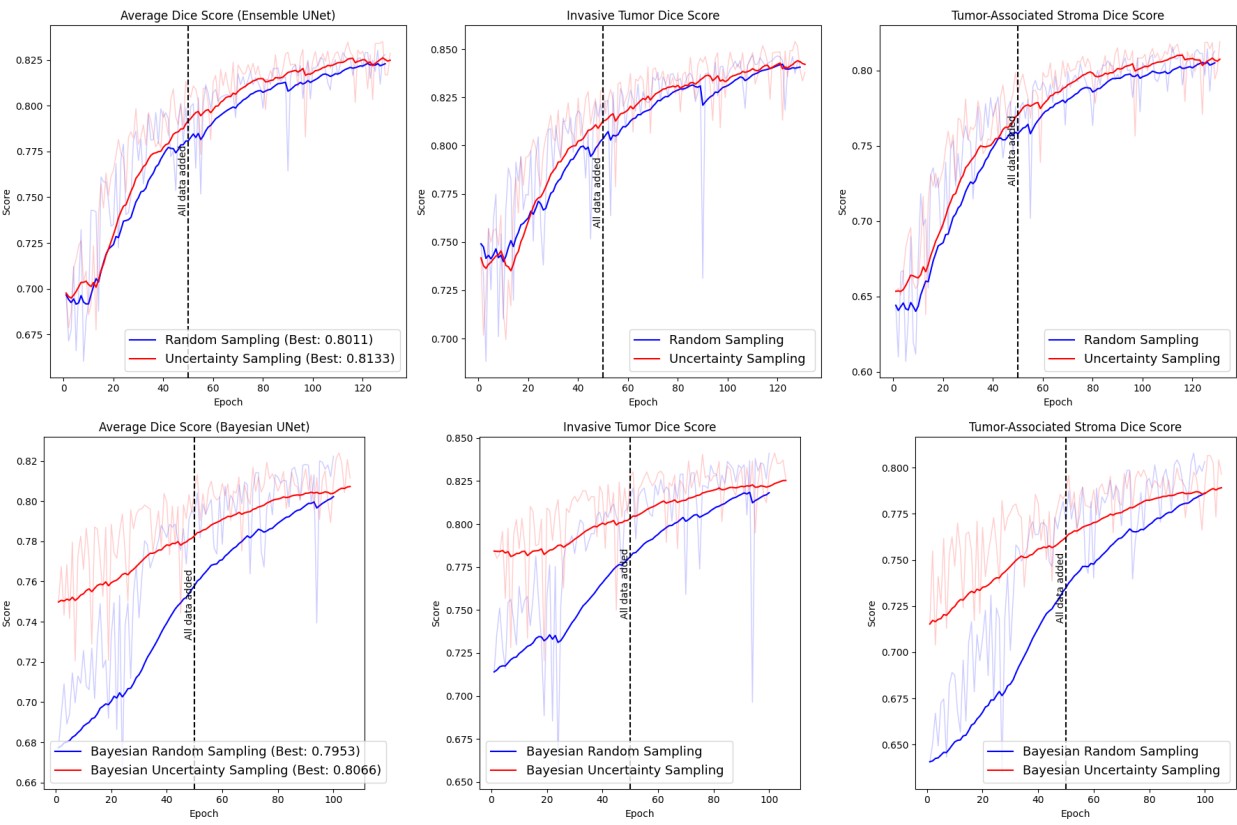

Figure 2: **Dice learning-curves on TIGER** Dice is tracked for two probabilistic architectures—the Ensemble UNet (top) and the Bayesian UNet (bottom)—under active learning with either red(uncertainty sampling) or blue (random sampling )faint lines show the raw per-epoch scores, bold lines a exponential moving average. The vertical dashed line marks the point at which all data from data pool have been annotated and added to the training set.

## 4.2 CAMELYON17

We followed by the proposed method described in the Method section. We train pretrained ensemble nn-UNets and BayesUNe model on 311 image patches pre-training dataset, we evaluate the model on validation dataset with 2806 image patches. After the pretraining, we implement the active learning approach we proposed to iteratively select the uncertain sample from a dataset pool with 7279 image patches.

From the table 4.2 Both models begin from virtually identical pretrained baselines (0.923–0.926 Dice). Adding only 25 % of the most uncertain images propels the Bayesian UNet to 0.9576 Dice—an improvement of +3.5 pp over its baseline and around +0.6 pp over the Bayesian random-sampling run (0.9516). The edge persists at 50 % data (0.962 vs 0.956, +0.6 pp) and remains at 100% data (0.965 vs 0.960). The Ensemble UNet also profits from uncertainty selection (0.9523 / 0.9538 / 0.9600), but its gains over random sampling are smaller— around 0.5 pp at 25 % and 50 %, and 0.4 pp at 100% data— highlighting that Bayesian predictive uncertainty is the more

| Model | Pretrained | 25% added | 50% added | 100% added |
|---|---|---|---|---|
| Bayesian UNet (Uncertainty Sampling) | 0.923 | 0.958 | 0.962 | 0.965 |
| Bayesian UNet (Random Sampling) | 0.923 | 0.952 | 0.956 | 0.960 |
| Ensemble UNet (Uncertainty Sampling) | 0.926 | 0.952 | 0.954 | 0.960 |
| Ensemble UNet (Random Sampling) | 0.926 | 0.947 | 0.950 | 0.958 |

Table 2: **Dice segmentation scores on CAMELYON17** The model performance of the pretraining model and after we added 25%, 50%, and 100% of the remaining pool. Uncertainty sampling lets both architectures reach 0.96 Dice with only 25%–50% extra data, while random sampling needs the full dataset to approach the same performance.

informative query signal for this task as shown by a segmentation instance from Figure 3

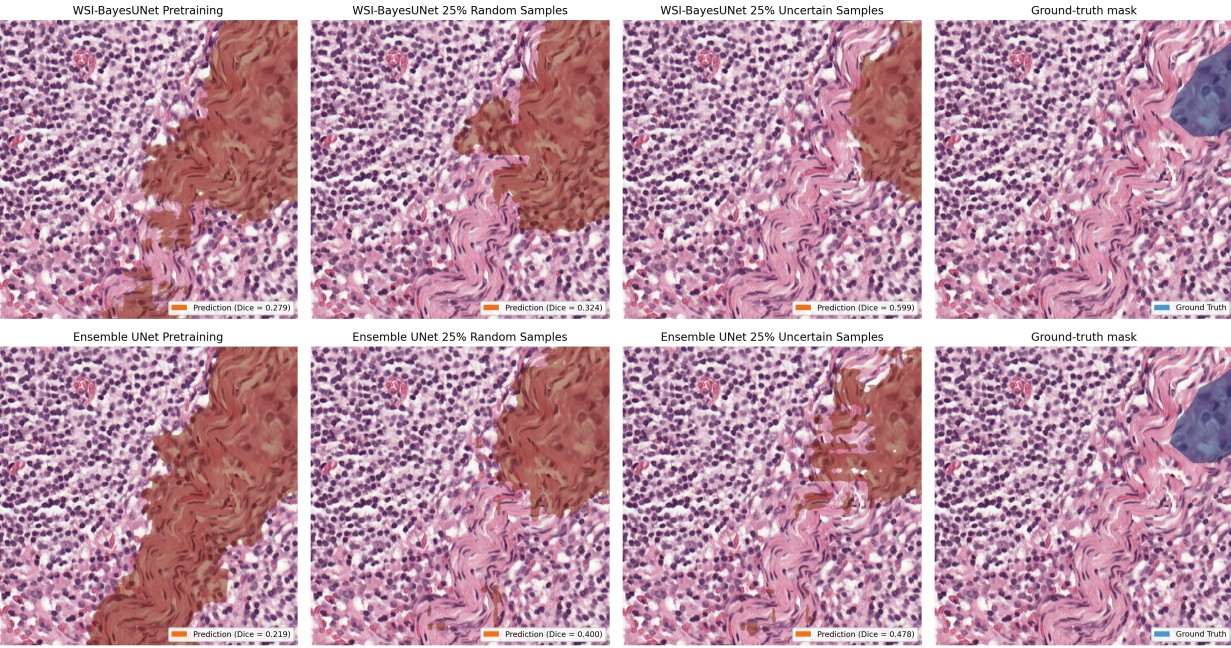

Figure 3: **Segmentation Instance from CAMELYON17** Our WSI-BayesUNet improved the pretrained model on a area that are difficult to segment. The WSI-BayesUNet uncertainty sampling improved pretraining model by 115pp while random sampling only improve the pretraining model by 16pp. Although Ensemble UNet uncertainty sampling improved the pretraining model by 118pp, the random sampling improved the pretraining model by 82pp as well

Overall, uncertainty sampling lets both architectures surpass 0.96 Dice with only half of the annotations, whereas their random-sampling counterparts require the entire dataset to reach comparable performance. From Figure 4, Uncertainty selection boosts both models, but the gain is clearly

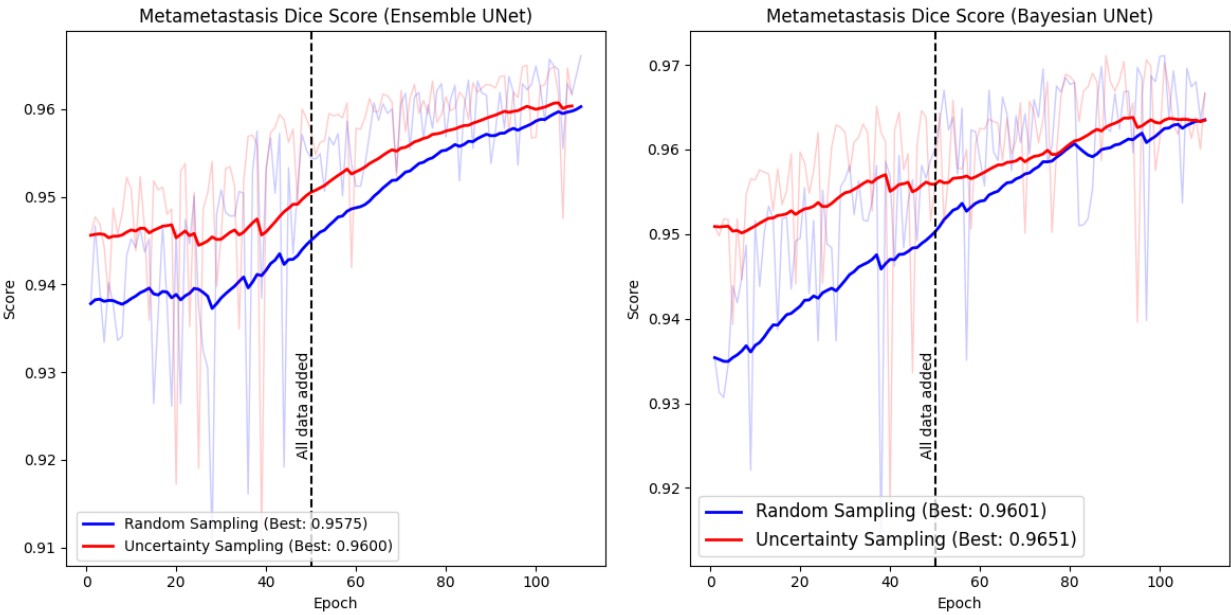

Figure 4: **Dice learning-curves on CAMELYON** the Ensemble UNet (top) and the Bayesian UNet (bottom)

larger and more sustained for the Bayesian UNet. Random sampling never closes the gap for either architecture, which validates the results from the TIGER dataset.

## 5 Conclusion

Our study showed that Bayesian U-Net effectively overcomes key disadvantages inherent to ensemble-based U-Net methods for predictive uncertainty quantification in histopathological image segmentation. The Bayesian approach consistently provided more robust, theoretically grounded, and computationally efficient uncertainty estimates. Experimental results from the TIGER and CAMELYON17 datasets confirm that Bayesian uncertainty-driven active learning substantially outperforms random sampling and ensemble-based uncertainty, achieving higher Dice scores with significantly fewer annotated samples. This demonstrates that Bayesian predictive uncertainty effectively guides model training and annotation prioritization, enhancing data efficiency and clinical applicability.

Future research directions include exploring advanced Bayesian approximation methods to reduce computational overhead further as Bayesian model training involves the representation of the network's weights distribution, integrating uncertainty quantification into larger-scale clinical workflows.

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
