# OpenReview forum: "WSI-BayesUNet: Uncertainty-Aware Deep Learning for Histopathological Image Segmentation with Active Learning"
_MICCAI.org/2025/Workshop/COMPAYL — COMPAYL 2025_

### Official Review · Reviewer_ujSk · 2025-07-09
**This study proposes a Bayesian U-Net framework for histopathological image segmentation that combines principled uncertainty quantification through variational inference with active learning, leading to improved segmentation accuracy, reduced annotation effort, and faster convergence compared to traditional ensemble methods.**

**Rating:** 4
**Confidence:** 4

**Review:**

**Pros:**

1- The paper proposes a principled Bayesian U-Net framework for histopathological image segmentation, which addresses the challenge of uncertainty estimation more efficiently than traditional ensemble methods.

2- The integration of active learning is well-motivated and demonstrates clear benefits in reducing annotation effort while maintaining or improving accuracy.

3- Experimental results on two benchmark datasets (TIGER and CAMELYON17) show competitive comparisons in terms of performance.

**Cons / suggestions for improvement:**

1- A block diagram or visual representation of the proposed methodology, particularly how the ensemble was performed, would greatly enhance clarity and reproducibility.

2- Please ensure that the percentage improvements reported in the tables are matches in the text details.

3- Since the evaluation is done at the tile level, it would be helpful to discuss the effect of whole-slide aggregation. Specifically, does BayesUNet require any additional post-processing steps for whole-slide inference?

4- Effective ensemble methods typically benefit from diversity among models. To that end, it is important to specify what variations (e.g., different augmentations, hyperparameters, etc.) were introduced to make each ensemble branch learn differently. What exactly was changed in each branch to ensure this diversity?

---

### Official Review · Reviewer_xgxN · 2025-07-15
**WSI-BayesUNet: Uncertainty-Aware Deep Learning for Histopathological Image Segmentation with Active Learning**

**Rating:** 3
**Confidence:** 4

**Review:**

The overall work is sound and addresses an important topic.

-The citation format in the whole document should be revised. For instance, Van der Laak et al. (2021) should be (Van der Laak et al., 2021).

-" 4,290 image patches, with 214 patches used for pre-training, 1,930 held out for validation, and the remaining 2,146 " - is the split slide-aware? Is it possible to have patches from the same slide in training and testing? If so, the results may optimistically estimate the performance.

-The SoTA on uncertainty estimation is not strongly presented. There is plenty of work on Bayesian uncertainty estimation.

-"with prior distributions p(w | D)" -> posterior distribution. There are several other typos in the document.

-Why only 5 models in the ensemble, and also only five sampled models? It seems unnecessarily short for a robust estimation of uncertainty. The conclusions could be very different if more models were included in the comparison.

---

### Official Review · Reviewer_n8PX · 2025-07-16
**This paper presents a Bayesian U-Net with active learning for histopathology segmentation, requires clearer validation of computational efficiency, simulated annotation, and hyperparameters analysis to fully support its claims. (Boardline))**

**Rating:** 3
**Confidence:** 4

**Review:**

The paper presents a Bayesian U-Net framework for uncertainty quantification in histopathological image segmentation, integrating active learning to improve annotation efficiency. The approach is well-motivated, and the results on TIGER and CAMELYON17 datasets demonstrate promising improvements over ensemble methods. However, several methodological and experimental concerns need clarification.

Strengths:
1. Novel integration of Bayesian U-Net and active learning for histopathology.
2. Clear empirical gains over random sampling and ensembles in data efficiency.

Major Concerns:
1. The paper claims Bayesian U-Net is "computationally more efficient" than ensemble methods but lacks detailed comparisons (e.g., training time, memory usage, or hardware specifications). Variational inference (VI) can be costly due to doubled parameters (mean + variance per weight). How does the computational cost scale with model size or dataset complexity?
2. The active learning loop uses "simulated annotation" with existing labels, which may not reflect real-world annotation dynamics (e.g., pathologist variability or label noise). How robust is the method to imperfect annotations?
3. The ensemble baseline uses five U-Nets, but the Bayesian U-Net uses five Monte Carlo samples. Are these comparable in terms of parameter count or FLOPs? A fair comparison should match computational budgets.

Minor Concerns:
1. No details on VI-specific hyperparameters (e.g., prior scales, KL weight annealing).
2. The "extended U-Net" for ensembles is mentioned but not described (e.g., skip connections, depth).